# Virtual Reality and Spatial Augmented Reality for Social Inclusion: The "Includiamoci" Project

**Valerio De Luca** [1], **Carola Gatto** [2], **Silvia Liaci** [3], **Laura Corchia** [2], **Sofia Chiarello** [1], **Federica Faggiano** [1], **Giada Sumerano** [1] and **Lucio Tommaso De Paolis** [1,*]

1  Department of Engineering for Innovation, University of Salento, 73100 Lecce, Italy
2  Department of Cultural Heritage, University of Salento, 73100 Lecce, Italy
3  Department of European and Mediterranean Cultures, University of Basilicata, 75100 Matera, Italy
*  Correspondence: lucio.depaolis@unisalento.it; Tel.: +39-0832-297736

**Abstract:** Extended Reality (XR) technology represents an innovative tool to address the challenges of the present, as it allows for experimentation with new solutions in terms of content creation and its fruition by different types of users. The potential to modulate the experience based on the target audience's needs and the project's objectives makes XR suitable for creating new accessibility solutions. The "Includiamoci" project was carried out with the aim of creating workshops on social inclusion through the combination of art and technology. Specifically, the experimentation involved ten young people between the ages of 28 and 50, with cognitive disabilities, who participated in Extended Reality workshops and Art Therapy workshops. In the course of these activities, the outputs obtained were two: a virtual museum, populated by the participants' works, and a digital set design for a theatrical performance. Through two tests, one on user experience (UX) and one on the degree of well-being, the effectiveness of the entire project was evaluated. In conclusion, the project demonstrated how the adopted solutions were appropriate to the objectives, increasing our knowledge of UX for a target audience with specific user needs and using XR in the context of social inclusion.

**Keywords:** social inclusion; extended reality; well-being; virtual museum; user experience; spatial presence



## 1. Introduction

Today, in the social field, the shift from a culture of integration to a culture of inclusion is aimed at enhancing the diversity of the individual to elevate the empowerment of the group. The "Includiamoci" project borrowed its name from this complex vision and involved 21 actual participants, 10 of whom constitute the experimental sample. The sample is aged between 28 and 50 and is composed of special needs people, with intellectual and/or relational disabilities. The project lasted six months and led to the implementation of social inclusion workshops through visual art, performance art, and technologies. The project, which was briefly introduced in an earlier preliminary paper [1], is presented here in a more extended form that includes more technical details on the application development (in particular on the implementation of a video mapping experience for the theater) as well as a user experience study. The adopted methodology was based on the combination of Art Therapy and eXtended Reality technologies, designed to stimulate participants and engage them in a process of self-expression and creative storytelling [2].

Extended reality (XR) is an umbrella term for computer-generated environments that merge the physical and virtual worlds or create an entirely virtual experience for users. XR encompasses Augmented Reality (AR), Virtual Reality (VR), and Mixed Reality (MR) [3].

In a Virtual Reality experience, users are fully immersed in a simulated digital environment. The gaming and entertainment industry were early adopters of this technology; however, companies in several industries are finding VR to be very useful.

In Augmented Reality, virtual information and objects are overlaid on the real world. This experience enhances the real world with digital details such as images, text, and animation. In contrast to virtual reality, users are not isolated from the real world. You can access the experience through AR glasses or via screens, tablets, and smartphones.

In Mixed Reality, digital and real-world objects coexist, and the user, wearing an MR helmet, can interact with them in real time. Many companies are using mixed reality to work to solve problems, support initiatives, and make their businesses better.

By bringing all this together, XR can uncover a broad new spectrum of opportunities across real and virtual-based environments.

The project participants, coordinated by the tutors, worked on the creation of artistic content and its fruition through Virtual Reality and Spatial Augmented Reality technologies, thus effectively acting on the capacity for self-expression and sense of self-worth. The decision to combine technologies with Art Therapy was dictated by several benefits, since through the digital medium, the artistic process is emphasized in its nonverbal language component, and comes to be enjoyed effectively and horizontally, allowing participants to express feelings and emotions and see them represented and communicated to all.

The project outputs were mainly two:

- The implementation of a virtual museum intended to house the works of art created by the participants; they can navigate and enjoy their works through a Virtual Reality (VR) helmet and a tablet device;
- A theatrical performance created entirely by the working group, enriched by virtual sets through the use of the Spatial Augmented Reality (SAR) technique.

Through two tests, one on user experience and one on the degree of well-being, the effectiveness of the entire project was evaluated, the results of which are described in this paper.

The project is carried out thanks to a strategic partnership that involves the Augmented and Virtual Reality Lab (AVR Lab) of the Department of Innovation Engineering (University of Salento), the Astragali Teatro theater company of Lecce, and the "NovaVita Elena Fattizzo" Association (Casarano, Lecce).

The rest of the paper is structured as follows: Section 2 presents a brief survey of the related work; Section 3 describes the "Includiamoci" project; Section 4 describes the development of a virtual reality environment for Art Therapy; Section 5 describes the design of a video mapping projection for Art Therapy; Section 6 presents the evaluation of the user experience; Section 7 reports the conclusions.

## 2. Related Work

When Virtual Reality is associated with the word museum, it is often referred to as a virtual museum. At the time of its introduction in the early 1980s, a virtual (or digital) museum was a digital representation of the collections of a physically existing museum [4]. This type of realization met a twofold need. First, it offered curators an excellent solution for both storing and consulting documentation about the collections and for handling administrative procedures (restoration, loans, etc.). The first configuration of the virtual museum coincided with the digitization of the traditional inventory and catalog. At the same time, it was realized how this technology can also be used to improve the effectiveness of communication strategies, encouraging a more conscious use of museum content.

A clear definition of a virtual museum is that provided by V-MUST (Virtual Museum Transnational Network), a European Union-funded network of excellence that aims to provide the heritage sector with the tools and support to develop virtual museums that are educational, entertaining, enduring, and sustainable. According to this definition [5], a virtual museum is a digital entity that draws on the characteristics of a museum, in order to complement, enhance, or augment the museum experience through personalization, interactivity, and richness of content. Virtual museums can perform as the digital footprint of a physical museum, or can act independently, while maintaining the authoritative status as bestowed by ICOM in its definition of a museum [6].

Virtual Reality, when accompanied by sound storytelling that starts from the study of the museum collection, enables the creation of immersive, interactive and collaborative virtual environments, can be a decisive innovation paradigm both in a time of emergency and in normal life, capable of coping with the most diverse narrative needs.

A systematic review can be found in [7]. Here, the state of the art up to 2018 of AR, VR, and MR systems as a whole and from the perspective of cultural heritage is analyzed, identifying specific areas of application and their purpose, as well as technical specifics related to visualization and interaction modes. In conclusion, the authors emphasized that augmented reality is preferable for the enhancement of exhibitions. Similarly, virtual reality seems better for virtual museums, and mixed reality more feasible for both indoor and outdoor reconstruction applications.

The first multi-user virtual archaeological museum in Europe was created in 2008 and is called the *Virtual Museum of Ancient Flaminia* [8], available online or inside the Roman National Museum of Terme di Diocleziano in Rome.

Another experiment was conducted in 2017 at the MACRO in Rome, where it was possible to relive the experience of visiting the exhibition *From Today to Tomorrow: 24 h in contemporary art*, which ended in the previous year. Using Oculus Rift visors, it was possible to move through the rooms, walk around the works, and rediscover an exhibition that no longer exists.

In [9], a VR application for the Archaeological and Naturalistic Park of Santa Maria d'Agnano in Ostuni has been described. The application gives the user the possibility to explore the Upper Paleolithic settlement and interact with the old manufacts.

However, digital technologies are often used as tools to facilitate accessibility processes [10]. Accessibility is the basis of inclusion, as it sets up the means necessary for it to become possible, acting, among other things, on overcoming physical barriers. VR allows users to immerse themselves in places of natural and cultural interest, establishing contact with them that would otherwise be impossible in some circumstances. Thanks to specific devices, the user is dropped into an environment that allows new sensory experiences, moving and interacting in a space that simulates an enriched reality free of physical obstacles.

This is the idea on the basis of Masseria Torcito Experience, an application developed to put everyone in a position to enjoy a structurally inaccessible environment, that of an underground oil mill in Salento. The 3D model is fully navigable and has been enriched with 3D content illustrating the operation of the machines existing in the mill in order to satisfy the curiosity of a wide and heterogeneous audience [11].

Another application was developed to provide users with the possibility of a virtual exploration of an inaccessible ancient castle in Corsano, a small town in Italy, with historical information about it [12].

Another area of application in the cultural field is the performing arts. Among the possible approaches, video mapping, a special form of Augmented Reality, can transform any surface, flat or irregular, into a dynamic surface capable of enriching human sensory perception. Video mapping projections can become a means of creating virtual stage sets, animating walls, and creating storytelling that can be enjoyed by several people at once, generating an exciting collective experience.

In [13], the authors showed how video mapping, beyond its purely technological aspect, can be linked to cultural heritage and represents a tool that can become a mediator of culture, tradition, and legends. Another approach for the performing arts is reported in [14], where the authors described the implementation of an innovative ballet with AR elements and elaborated on the methods of collaboration and the main challenges encountered. The project presented in [15] describes a similar experience of a collaborative project using AR to allow dancers to interact with virtual content while performing on stage. The project quickly attracted the interest of students and professors, as the arts and technology contributed differently to the project as production was driven by artistic vision, while science provided a service to art in the form of new technology.

This is the context for the "Includiamoci" project, which was born out of the need to make the same XR technologies usable by a target audience with psycho-physical disabilities as well, overcoming not only physical but also virtual architectural barriers that these users often face. In fact, by exploiting VR and AR technologies, it will be possible respectively to make users enjoy an immersive experience independently and to stage a show where the users themselves are protagonists through the self-produced digital scenographies. Drawing partly on work focusing on *Spatial Learning* [16], *Spatial Presence* [17], and *Spatial Memory* [18] in XR environments, a questionnaire was also adopted to assess the *Spatial Presence* of disabled users in a virtual environment in which they had the opportunity to recognize reconstructions of artefacts they had produced themselves.

## 3. The "Includiamoci" Project

### 3.1. The Social Inclusion Workshops

The idea of starting a social inclusion hub stems from the need to create spaces for inclusion and mutual exchange, where everyone could feel welcomed and valued in their human and creative potential.

Art is the common thread of the proposed experience: art as a form of personal and individual expression, art that should allow each person to express feelings and emotions by resorting to a language that is not necessarily verbal.

The social inclusion hub of the "Includiamoci" project was designed to allow participants to express themselves freely, giving everyone the opportunity to communicate with themselves and with the work group, in a path that enhances the personality of the individual.

The activities carried out were based on the methodology of Art Therapy, which, as Edwards argues, concerns both the process of creating images (from the crudest doodles to more sophisticated art forms) and the establishment of a relationship between therapist and patient, with the aim of allowing the latter to share and explore emotional difficulties or to use a simple crayon to develop and shape hitherto unexpressed feelings [19]. Art Therapy is in fact a form of communication that does not necessarily have to make use of speech to allow the individual to express feelings and emotions, embarking on a path that can lead to his or her transformation, evolution, and growth.

Several studies have shown that this practice can significantly improve the lives of people facing physical and/or psychological distress or social isolation by relieving stress, encouraging nonverbal communication, and strengthening self-esteem [20].

With this in mind, the social inclusion hub is configured as spaces for socializing and encouraging creativity, with the aim of bringing out the strengths of each participant and allowing them to tell their story through artistic expression in all its forms.

Therefore, during the weekly meetings, space was given to various forms of artistic expression: drawing, creation of small objects with modeling clay or compositions with collage, music, and theatrical performance.

### 3.2. Working Methodology

Coordinated by the "NovaVita Elena Fattizzo" Association (Casarano, Italy) with the collaboration of AVR Lab of the Department of Innovation Engineering, University of Salento (Italy) and the Astragali Teatro company of Lecce (Italy), the project involved 21 people who are between the ages of 28 and 50 and have an intellectual and/or relational disability.

The project had three main objectives:

- To implement social inclusion workshops based on Art Therapy workers in order to foster the creativity of each participant and to create an environment in which each person could feel included and comfortable;
- To implement a virtual museum intended to house the graphic and sculptural works created during the weekly meetings and that can ensure that they can be enjoyed in an environment designed according to the needs of the end users;

- To bring to the stage a play performed by the working group and whose sets were created through the use of Spatial Augmented Reality techniques.

### 3.3. The Art Therapy Lab

The weekly meetings of the Art Therapy hub took place working in concert with the educators on duty at the "NovaVita Elena Fattizzo" Association.

The first phase was focused on getting to know the group, in order to create a relationship of trust between the participants and the working team and to foster the acclimatization of each one within entirely new spaces, such as those of the university.

The initial dialogue and interaction phase, aimed mainly at getting to know each other, was followed by a phase of introduction and gradual approach of the participants to new technologies, through immersion in virtual reality environments by means of wearable devices, such as cardboards and 3D viewers.

The third step involved the Art Therapy phase properly. Participants were asked to produce graphic works based on themes that, from time to time, were proposed by the project team. This phase allowed each participant to make use of colors and modeling clay to tell his or her story, aspirations and passions, and to foster the creation of an atmosphere marked by increasing confidence with peers and tutors (Figure 1).

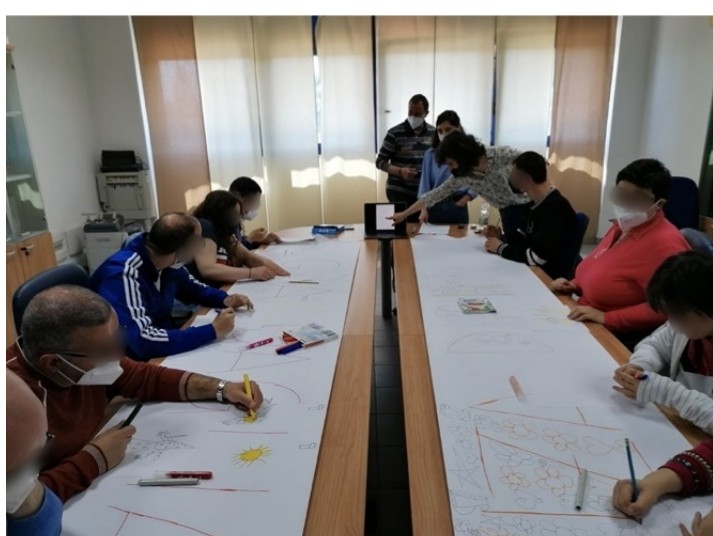

**Figure 1.** The Art Therapy Lab.

One day was devoted to the music workshop in which the Kodaly Method [21] was used, working with the body and voice in a synergistic and collaborative manner. Music, through its educational, didactic and experiential power, made it possible to create a real connection between one subject and another through the body and voice of each. Each identity sound is correlated with others, and the choral activity led to knowledge of self and others.

The works created during the meetings flowed in the fourth phase into a virtual museum, usable by means of a VR visor. The virtual museum setting was designed with the specific target audience of end users in mind, with environments that are easy to explore and emotionally engaging.

In the virtual museum, the four exhibition rooms and the "cinema room" (intended for the screening of a film starring the participants and featuring small video interviews) can be explored independently or, in the case of people with greater interaction difficulties, with the help of an operator managing the controllers.

The fifth phase involved the preparation of the performance and sets for the theatrical performance that closed the experience and which, under the direction of the Astragali Teatro company from Lecce, staged excerpts from *Il libro degli errori* by the writer Gianni Rodari (1920–1980).

### 3.4. The Role of Technology

During the weekly meetings, participants received a gradual introduction to eXtended Reality in order to take advantage of Virtual Reality, Augmented Reality and Mixed Reality to modify reality.

In recent years, these technologies have been gaining increasing interest, and several experiments are targeting the world of disability in an attempt to address complex problems and offer viable alternatives to the lack of certain abilities. For example, placing subjects within scenarios that are difficult to actually recreate and subjecting them to certain stimuli with therapeutic intent, digitally correcting patients' senses and perceptions, and enabling physically disabled individuals to interact with the machine and explore environments that, in reality, would be difficult to visit.

Specifically, Virtual Reality-based scenarios (passive, exploratory, and interactive) to allow users to hear and see what is happening in the virtual environment, move by taking advantage of different modes (walking, jumping, flying, or swimming), and interact with the objects and characters present, giving them back the feeling of being truly immersed in it.

Users with disabilities can thus immerse themselves in a virtual environment and engage in a range of activities that take place by means of a simulator free of the limitations that their disability imposes, and they can do so safely. In addition, the knowledge and skills that are gained within the virtual environment can be transferred to the real world [22].

In light of these considerations, the work team chose to implement a virtual museum intended to house the works produced during the Art Therapy workshop taking into account the special needs of the end users. The User Experience (UX) design and implementation steps will be described in the following paragraphs.

## 4. From Art Therapy to the Virtual Museum

The social inclusion hub, which allowed the project participants to express themselves artistically and bring to life personal works of art on paper and in clay, was followed by the digitization of these artworks in order to place them in a virtual museum.

The museum experience, accessible both through Meta Quest 2 and on tablets, was customized to the target of users who were the protagonists of the activities, allowing them to get in touch with Virtual Reality both independently—where there are prerequisites—and with the support of a tutor.

Firstly, 3D models of the clay sculptures were made through the photogrammetric technique in order to generate digital statues to be displayed on stands with labels indicating the author's name.

Next, drawings were scanned to make them into paintings to be placed in various frames scattered throughout the exhibition space. On these same drawings, moreover, 2D animations of some elements were created, adding with special software specific sounds for each painting.

The same video editing tools were also used for the creation of a final video, which can be viewed in the Museum's cinema room, in which the participants described their creative process during the Art Therapy sessions.

Once the photogrammetric sessions were finished and the multimedia files were produced, the 3D environment was created in which to display the works produced by the users during the Art Therapy laboratory. The choice turned to museum space, by nature an exhibition container, but also because it is increasingly established how museums can benefit health and help develop positive well-being and resilience [23].

Several factors were taken into account when modeling the museum, including, in the first instance, the heterogeneous target audience in terms of age, gender, and psycho-physical disabilities. Therefore, a relaxing environment with warm colors and soft lines was created so that users could enjoy it without emotional overexertion.

Once all the elements useful for the creation of the virtual museum were gathered, we moved on to its implementation in Unity, a graphics engine capable of generating fully or partially immersive virtual environments. Unity is a cross-platform game engine that has been gradually extended to support a variety of desktop, mobile, console, and virtual reality platforms. The engine can be used to create three-dimensional and two-dimensional games, as well as interactive simulations and other experiences [24]. The engine has been adopted by industries outside video gaming, such as film, automotive, architecture, engineering, and construction. Unity Asset Store includes 3D and 2D assets and environments for developers to buy and sell.

The implementation work started from a careful study of the needs: given the heterogeneity of the subjects involved, it was necessary to create a virtual environment that could be easily visited and enjoyed, both by the direct beneficiaries of the project and by their tutors, who were assigned the task of guiding or supervising—while the user wears the visor or uses the tablet—the visit to the virtual museum.

### 4.1. Photogrammetry of Sculptures

The clay sculptures made during the Art Therapy sessions were digitized in 3D through photogrammetry. This technique can generate, from the union of several photos, three-dimensional models of small, medium, and large objects [25]. Starting with a thorough photo session, the result of which must be a series of photos that overlap with each other to a small degree, we then proceeded to associate commonalities among the photos. This was made possible by using Agisoft Metashape [26], a software that can detect the points of overlap between one photographic shot and another in the variations of gait and colour of the object. From that process, a point cloud was created from which a mesh (i.e., a 3D model) with low, medium or high resolution textures can be generated.

In accordance with what has just been said, the first phase was the photo shoot of ten clay works. This process required a place that had even light and a neutral background on which to place the sculptures. Once the set was arranged, the camera was set to the specifications best suited for an optimal result: in particular, manual shutter release and fixed ISO (with a value at 1600) were set so that there would be no variation in the intensity of colors and lights; the other parameters set were focal length at 34, F-stop at F/7.1, and Shutter at 1/100. In order to have the maximum resolution of the elements in the photos, it was chosen to shoot in .RAW format. Depending on the complexity of the sculptures, the number of photos taken is correspondingly increased or decreased.

The photos were edited using the Camera Raw plug-in of Adobe Photoshop 2022 to achieve greater uniformity: this allowed the parameters of exposure, contrast, temperature and hue to be set similarly for all photographic shots. Ultimately, the files were exported in .TIF format (with 16-bit color depth per channel) so that they could be processed by Agisoft Metashape at maximum performance.

Once all the files were exported, they were imported to the photogrammetry software and analyzed to understand the quality of the photos: from the result of this analysis, all files of less than 0.5 quality were eliminated to avoid processing errors.

Thanks to the alignment of the photos, the baseline for the 3D model of the clay sculptures was created (Figure 2). The first output of this procedure is a set of Tie Points, from which a Depth Map of medium quality can be generated by using the Aggressive filter. Then, the process moves on to generate the Dense Cloud of high or medium quality, depending on the complexity of the sculpture: Medium quality was selected for simple sculptures; High quality was selected for more complex sculptures. In all cases, the Mild parameter was selected, which is useful for obtaining a good level of detail.

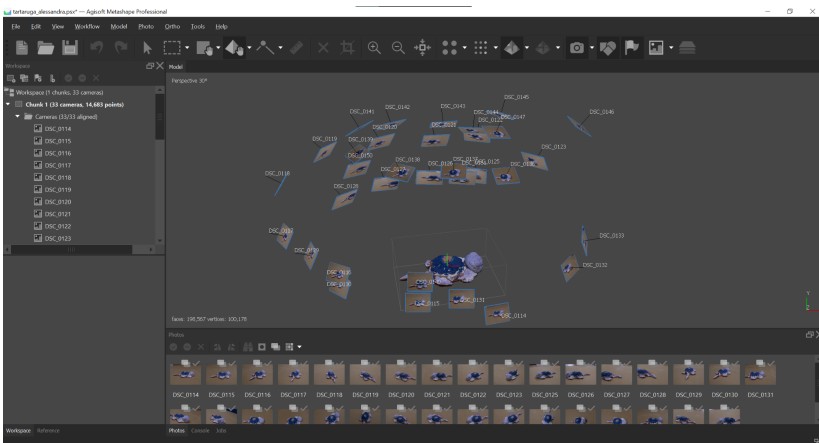

**Figure 2.** Camera alignment process in Agisoft Metashape.

Once these elements were obtained, it was possible to generate the 3D mesh (Figure 3), which was then cleaned, removing what did not serve the purposes of the model, in particular the background. A mosaic texture was then generated for each 3D model.

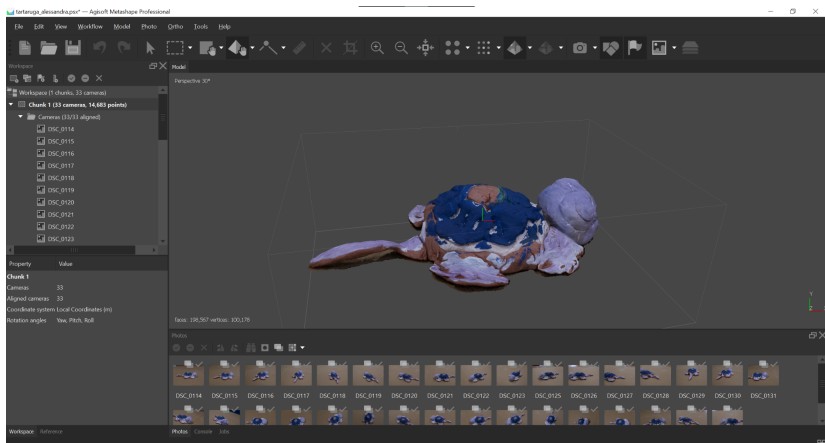

**Figure 3.** Creation of the 3D mesh of a clay sculpture in Agisoft Metashape.

The 3D models were exported in .fbx format to be imported into Unity for the creation of the virtual museum.

*4.2. Multimedia Content Development*

During the Art Therapy workshops, participants, in addition to the creation of clay sculptures, had the opportunity to express their creativity through the creation of drawings with colored markers on paper sheets. Each of them gave free play to their imagination by making subjects that were later digitized and animated in the post-production phase. This phase involved the use of specific software such as Adobe Photoshop 2022 and Adobe After Effects 2022, the main applications for processing digital images and graphic animations.

For the digitalization, top-down photographs of the drawings were taken, which were then imported into Photoshop, subdivided and cut into the necessary components to be consequently animated: using the quick selection tool, individual portions were carefully selected and contoured, and with the clone stamp, the spaces left empty by the separated elements were filled in. The operation consists of copying the pixels of a given portion of the image and duplicating them in the area you wish to occlude. This generates two layers: the background and the cut-out element that will be animated.

Once the work in Photoshop was finished, the individual images with their respective cutout elements were imported into After Effects for the animation process. This computer graphics and video editing software can handle 2D and 3D layers and customized effects,

within specific compositions, which represent the main structure of a clip. Each composition has its own Timeline within which the properties of each layer can be animated using Keyframes. Having imported the contoured images, simple and short animations were created, in order to later upload them to the Unity platform and provide them with background music for the purpose of greater user engagement.

A dedicated space was created inside the virtual museum, giving the user the opportunity to enter a cinema hall and view the concluding footage of the workshop experience. Such footage was developed in order to give uniformity to the content of the virtual museum: it includes a sequence of photographs taken of participants as they show their drawings on camera and video content showing interviews filmed during the Art Therapy workshops. For this purpose, a set was specifically created with a workstation and lights to give more authenticity to the environment and the experience: following the same logic for uploading the drawings and photogrammetries in Unity, also in this case, it was necessary to develop a video that did not weigh down the final model in the platform; so, the editing phase of the individual sequences was very important, accompanied finally by a background music (the Instrumental of *Viva la Vida—Coldplay*). The video, therefore, begins with a film-style countdown that launches the interviews and ends with a collage of photographs culminating around the "Includiamoci" project logo.

The video editing software used was Adobe Premiere Pro 2022, which gave the ability to easily manage the different transitions between clips and audio.

*4.3. 3D Modeling*

In parallel with the photogrammetric session involving the clay sculptures and the creation of the paintings' animations, a 3D model of a museum was selected from the online library "Sketchfab" (License: CC Attribution. Model by Elin Höhler: https://sketchfab.com/3d-models/vr-gallery-house-baked-8e3b280476eb4790af97e2abd3d9be59, Modified and adapted, accessed on 30 December 2022) to be later customized on the 3D modeling software Blender.

The main changes involved applying textures on the mesh and implementing the model with simpler furnishings consistent with the proposed experience.

The museum model consists of two large rooms that were planned to be used as a cinema hall and exhibition hall, respectively, along with two smaller spaces that serve as appendages to the museum environment (Figure 4).

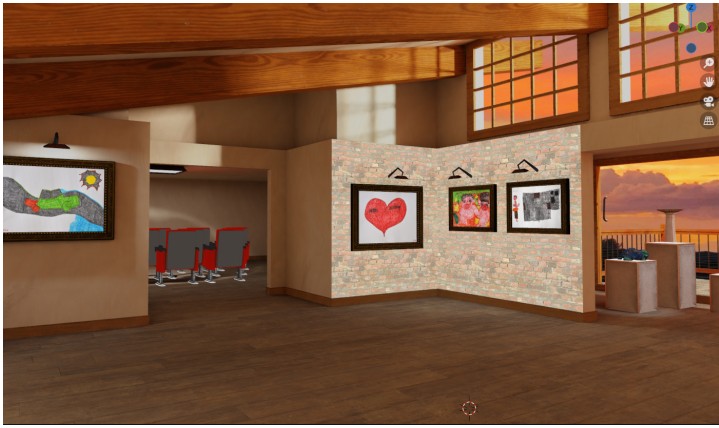

**Figure 4.** Visualization of the exhibition hall, cinema room, and balcony in Blender.

The light in the room comes from a balcony separated from the rest of the museum by a glass wall that, although not physically accessible in the application, gives a glimpse of the sky at sunset time. Two lit torches provide a frame that adds to the atmospheric view outside.

As for the furniture, ten pedestals of different heights were modeled on which clay sculpture models obtained by photogrammetric acquisition were prepared.

Similarly, the ten drawings made on paper, after being scanned, were placed on the museum walls inside frames molded in 3D from cubes.

At a later date, the names of the children-authors of the artworks were listed on a special label, placed below the associated paintings and sculptures.

A number of three-dimensional models of red armchairs arranged in three rows were imported into the cinema room in the direction of the white screen where, during the virtual experience, the user will be able to view a short film made from the interviews of Art Therapy workshop participants.

### 4.4. Implementation in Unity

The applications designed for the Virtual Museum "Includiamoci" were made usable through two different devices, with two different levels of immersion: in the former, the user is fully involved in the virtual experience by means of the Meta Quest 2 visor, while the latter involves the use of a tablet for a more simplified user experience.

In both cases, the development phases on Unity's Game Engine 2020.3.14f first concerned the definition of the user's movement in the virtual world, and then, the placement of the animated paintings and audio-visual content in the museum rooms.

### 4.4.1. The Immersive Application (DA QUI)

The immersive Virtual Reality application described in this paper was developed to be viewed using the Meta Quest 2 head-mounted display [27].

Meta Quest 2 is a stand-alone visor that offers the possibility to enjoy the virtual experience anywhere, without having to be connected to a PC via cable. In addition, the "Passthrough +" system ensures the user's freedom and safety of movement without colliding with real objects during the VR experience by setting the boundaries of the gaming area.

Once the device to be used was defined, the virtual environment was set up on Unity, starting with the configuration of the visor and controllers.

In particular, the first step involved the association of certain inputs (referring to rotation, body, and hand gestures of the user) to the Oculus Touch controllers of the Meta Quest 2 visor. This is done by means of the new "Action-based" input system, supported by the Open XR plug-in. The latter, defined as an open royalty-free standard, allows applications to be developed on a wide range of XR compatible devices, including the Meta Quest 2.

In order to continue, it was necessary to install two more packages:

1. "XR plug-in management" through which Meta Quest 2 controllers are identified as the object of the input actions to be configured;
2. "XR Interaction Toolkit" which includes the "default input actions", i.e., a set of preset actions already connected to individual controller buttons. However, in this project, some actions in the list have been modified by selecting from the "Bindings" a new action-button match to customize the user's movement in the virtual world, which has to take place through the teleport system.

In particular, two types of teleportation have been proposed, which complement the target audience of the VR experience:

- The user can move freely and autonomously around the virtual museum;
- The interaction with the controllers is considerably reduced, without hindering the full enjoyment of the VR environment for users with physical disabilities.

In both cases, twelve teleport stations have been set up in front of the exhibits (both paintings and clay sculptures).

The former movement mode takes advantage of the "Locomotion System", which manages the user's movement and rotation in the virtual environment and, by default, includes the "anchor" teleport system. The latter lets the user decide which of the twelve locations to move to, which, however, have spatial coordinates already preset by the

developers. During the game session, the user must point the controller at the circular icon located on the floor, which identifies the teleportation location to which he/she would like to move. At that point, the controller's ray turns white and, by clicking on the controller's grip button, movement takes place (Figure 5).

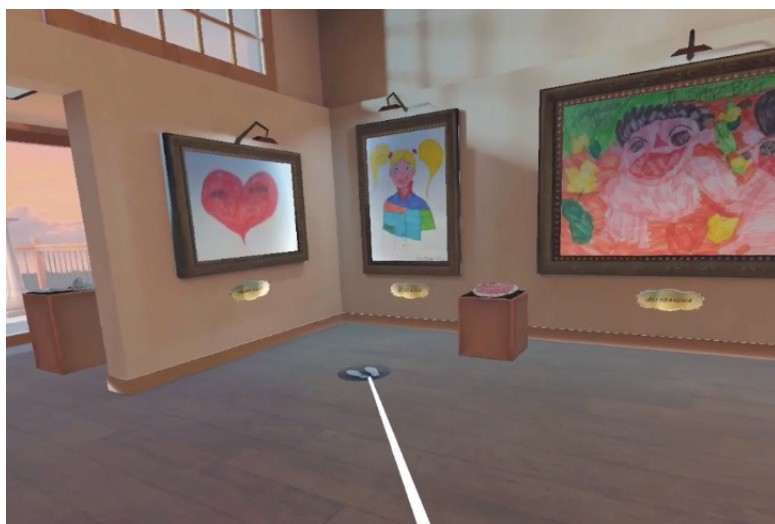

**Figure 5.** Virtual Museum navigation in Unity.

The second teleport mode was designed to facilitate fruition in the virtual environment by customizing the "Default Input Actions" mentioned above. Specifically, the user's teleportation movement was associated with a single button on the right controller. This choice was made in consideration of the higher usability of the right-hand controller. In fact, the educator, who accompanies the participants during the experience, by clicking on this button on the controller causes the user's movement in VR to take place. In this way, the user wearing the visor perceives the movement and, at the same time, the educator can monitor his/her own actions or those of the user (when it is deemed appropriate to have them interact independently) through the application's mirroring system in real time on any screen and on any device (smartphone, tablet, PC, and TV).

Next, within a script, it was set up that each time the button is clicked, a movement of the camera occurs, based on the spatial coordinates of the twelve previously declared teleportation locations.

### 4.4.2. Tablet Fruition of the Virtual Museum

To promote greater usability of the application, a simplified version of the "Includiamoci" application usable via tablet was designed, at the expense of a lower degree of user immersion and involvement in the virtual environment.

Although the model of the museum, with paintings and sculptures attached, is the same as that in the Oculus application, some changes occurred in order to implement the experience as a tablet application.

The choice of the tablet as the default device for this application is due to the better resolution than the smartphone in relation to the indicated target audience. For the development of the application, the Google VR SDK, generally employed for applications based on low-cost cardboard visors, was imported. However, by means of a special script acting on the display mode, the two lenses were unified. In this way, the stereoscopic and three-dimensional vision was lost, but the gyroscope and user movement mechanism imported with the Google VR SDK was preserved.

Specifically, another script handles the movement of the player within the virtual museum by acting on two elements: the direction of the camera raycast and the user's touch.

A force is applied to the registration of the user's touch, which allows the camera to move along the direction defined by its raycast, performing the motion.

The same mechanism is exploited for the display of the final video inside the cinema hall: when the raycast of the camera meets the collider applied to the cinema screen, the video is activated; as soon as the collision ends, the video pauses and resumes when the camera rotates back to the screen. This principle, however, was not applied for painting animations, which remain static, unlike the application in VR.

To prevent the user from unknowingly exiting the virtual world with a prolonged touch, colliders were set up all along the walls, museum furniture, and at the player, with which the camera was associated.

## 5. From Art Therapy to Spatial Augmented Reality for Theater

In the perspective of active and cooperative inclusion of this project, starting precisely from the concept of Art Therapy described above, among all technologies, the one that can offer a collective and meaningful experience without the use of auxiliary devices was chosen, which is Spatial Augmented Reality (SAR). The use of this technology made it possible to create a dynamic scenography of the theatrical performance. The decision to use digital scenography, starting with the graphic designs created during the workshop phase, allowed to characterize the show by making the entire experience unique, which started from the use of technologies as an asset to break down barriers against all kinds of discrimination. The theater became everyone's place and SAR technology enabled the uniqueness of the final product, the synthesis of a shared journey in which art and technological means created connections and bridged gaps.

Spatial Augmented Reality is a type of Augmented Reality better known as Video Mapping or Projection Mapping, which augments and enriches sensory perception through the addition and/or subtraction of information that allows physical reality to be perceived in a completely different way [28]. The main purpose of this technology is to create a dialogue, a hybridization, between real and virtual in a collective, shared experience.

This technology has the advantage of allowing the user to interact with reality without the use of accessory devices that could distract attention from the purpose of the realization and lead it back to the medium used to enjoy it. Through animated projection on the surface, it is possible to ensure greater participation of the audience with the surrounding space in which they are located. Through this technology, there is the possibility to transform any surface, flat or irregular, into a dynamic surface capable of enriching human sensory perception through the use of a computer and a video projector system [13]. Lights and shadows dress and animate the surface of objects, creating a balance between container and content, between physicality and virtuality.

The use of technology is very often linked to the projection of digital content onto vertical surfaces such as buildings and structures that, through visual art, are enhanced by spectacularizing events with ad hoc narratives. These events range from the first projection mapping experiment in the late 1960s at Disneyland for the *Haunted House* [29] to spectacular examples in recent years such as the 2011 projections on the Prague Clock, the interactive works on the Manhattan Bridge in 2012 [30], the project of a sustainable highway coming to life on the *De Rotterdam* skyscraper in the Dutch city (2013) [31]. The projections on the Guggenheim Museum in Bilbao in 2017 [32] are of great importance, as are several Video Mapping festivals around the world, such as the *Solid Light Festival* in Rome [33], the *iMapp Bucharest* in Bucharest [34] and the *Festival of Lights* in Berlin [35]. These examples concern the use of technology in outdoor spaces, where the enjoyment of places has changed dramatically. Visitors immerse themselves for a few minutes in new and exciting spaces that seemingly change their form but never their substance.

In the context of theater, however, the discussion becomes a bit more complex, since what the public enjoys is not only the theater intended as an asset that has a historical-artistic-cultural identity in itself, but the theater as content in which the performance is the primary object of this fruition. In recent decades, the theater has been an incubator of innovation, offering artists, performers, and directors ways and means to imagine their own performances. Audiovisual technologies have found fertile ground here, and SAR has

found important applications, such as Robert Lepage's landscape created by set designer Robert Fillion for the *Andersen Project* in 2005, a concave device that housed video projections that appeared to have a three-dimensionality and interact with the actor literally immersed in this virtual environment [36]. Another example was the perspective cage of *L'Ospite dei Motus* inspired by Pasolini's film *Teorema*, a monumental set design that hangs over and crushes the characters, consisting of a deep sloping platform closed on three sides composed of as many screens housing video images in an imposing video triptych that releases the illusion of a three-dimensional space, of an optical chamber, of a huge house without a fourth wall [37].

SAR technology follows a specific methodology that takes into account certain parameters such as:

- The optical deformations;
- The dimensions of the projection frame (height and width);
- The distance between projection frame and projector;
- The observer's point of view.

This detailed study leads to the selection of the most appropriate instrumentation for the specific context referred to. The calculations of Ansi Lumen, Aspect Ratio, Throw Ratio, and Resolution allow the selection of the most appropriate projector for the circumstances, taking into account the environmental lighting conditions.

*Projection Mapping and Virtual Scenography*

In the specific case of the project at hand, a digital scenography was created in the occasion of the theatrical performance *Il Paese senza errori* held on the 10 June 2022. This scenography accompanied the show with the choreographies of the participants based on some poems by Gianni Rodari.

The main objective was to create a digital scenic apparatus that best harmonized with the choreography and recited poems. The performance involved a 4:3 vertical projection screen, as a stage background, on which an adaptation of the projection content on the vertical surface was made (warping) [38].

Technical and narrative factors are necessarily linked, since each video projection in the theater follows the narrative thread and a synchronization between images, video, and sound. In addition, in the case of the "Includiamoci" project, it was crucial to follow the timing of the whole play so as to create a digital scenography in line with topics and timing.

Again, the same process of creating 2D animations was followed using Adobe Photoshop and Adobe After Effects computer graphics software. Depending on the theatrical content related to some of Gianni Rodari's poems, the participants again unleashed their creativity during the workshops, drawing and bringing to life a series of figures, letters and concepts found in the poems they would perform during the performance.

Concluding the digital scenic apparatus was a video in which the participants simulated a walk around the circumference of the spinning world, to  give emphasis and concreteness to the final poem *Girotondo intorno al mondo*. For this purpose, a recording day was organized in which participants were filmed walking on site above a green screen. The goal was to contour in post production their figures and edit them later so that they walk side by side around the world, which, in this case, is a drawing they made themselves.

The mapping work was done through the dedicated MadMapper 5.1.3 software [39], and a Christie LX500 XGA projector at 5000 AL and 1024 × 768 resolution in 4:3 projection ratio was used for projection.

## 6. User Experience Evaluation

After a testing session of the VR application (Figure 6), the user experience was evaluated in terms of both spatial presence and well-being by administering two different questionnaires as described in the following subsections.

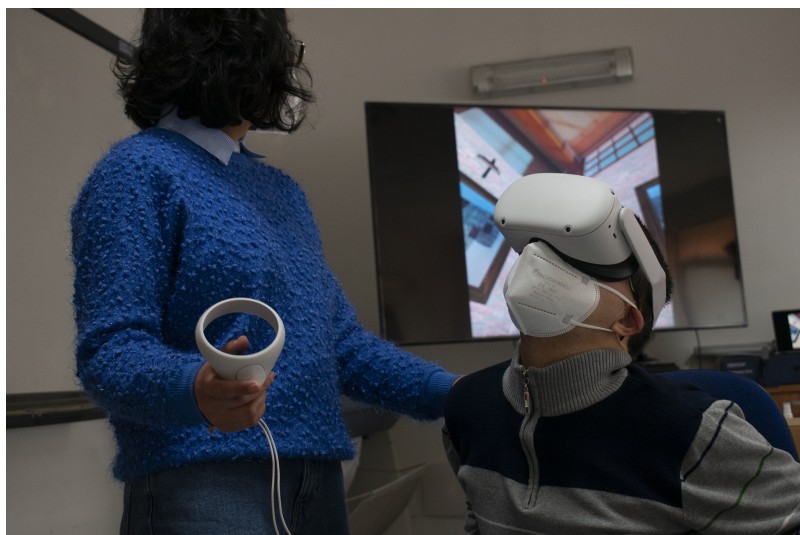

**Figure 6.** VR application testing.

*6.1. Spatial Presence Model*

The *Measurement, Effects, Conditions Spatial Presence questionnaire (MEC-SPQ)* [40], based on a two-level model of *Spatial Presence* [41], was chosen for the user experience evaluation.

The *self-location* component of *Spatial Presence* deals with the user's feeling of embodiment, i.e., the sense of being located in the virtual environment. This leads users to build a mental model of their bodies that also includes assumptions on the actions that they could perform in the virtual space: this phenomenon is described by a second component of *Spatial Presence* called *possible actions*. The *Spatial Situation Model*, a precondition for *Spatial Presence*, is a mental model [42] of the spatial environment based on the perceived spatial cues and on a user's spatial memories and cognition [43]. The devotion of mental capacities required to build the mental model is represented by *Attention allocation*, which can be *controlled* or even *involuntary*. *Controlled attention allocation* can be driven by motivational processes triggered by a match with a domain-specific user interest.

*Involvement* refers to the active and intensive processing of the environment, which includes thinking about, interpreting, elaborating, appraising, and assigning relevance to the content [44]. However, unlike *Spatial Presence*, *Involvement* does not imply a loss of mental contact of the user with the real world. In turn, *Presence* does not necessarily require active participation in terms of *Involvement*, but it can be a consequence of *Involvement*: a very high level of *Involvement* could lead the users to concentrate on the virtual content and not on the real environment. Distractions that may divert the user's attention from *Spatial Presence* may depend on technology (e.g., in the case of uncomfortable devices) or on the content (e.g., in the case of unrealistic narrative plots).

The analysis presented in [17] assessed the relationships among *Spatial Presence*, situational interest, and behavioral attitudes in a virtual museum navigation.

Several studies employed VR and AR to assess or assist *Spatial Learning*, which deals with the acquisition of spatial knowledge and the foundation of living interaction [16]. VR, in particular, is a surrogate for the real world to study how people learn the space in controlled laboratory conditions, where they can be protected against physical hazards. According to [16], *Spatial Learning* is influenced by spatial ability, cognitive workload, vision and hearing, cybersickness, and other factors (such as gender).

*Spatial Memory* is the part of human memory that handles information about the environment and influences orientation and navigation skills [45], which represent the ability to perceive and adapt the position of the body based on its relative position to other objects or environmental cues [46]. *Spatial Memory* can be seriously impaired by aging and neurodegenerative diseases, but it can be improved through virtual body representation,

spatial affordances, and virtual enactment [18]. Another study on visuospatial memory in navigational tasks [47] revealed that both older and younger users exhibit a better recall accuracy if the level of realism is adjusted by suppressing texture details from task-irrelevant elements of a virtual environment. A virtual environment to assess *Spatial Memory* was presented in [48]: experimental results showed the best performance in tasks where allocentric cues are used, where objects are represented primarily with reference to their spatial and configurational properties; moreover, a spatial strategy allows a faster completion of tasks. The same authors also highlighted the possibility of a future use of CAVE technology in new memory tasks to assess spatial pattern separation, pattern completion or *Spatial Memory* in elderly or Alzheimer's patients.

A recent study on spatial cognition involved children with physical disability in three spatial tasks [49]: the results showed lower performance in relation to the age of the children, but generally in line with their non-verbal mental age, with the exception of the mental rotation test, which proved particularly problematic for children in wheelchairs; unusual error patterns were noted in the spatial programming task for both wheelchair users and children with independent locomotion. Brain–Computer Interfaces (BCI) can be used for real-time assessment of spatial cognition in virtual environments, but there are some aspects to be improved [50], such as the combination of natural interaction in VR with BCI, the accuracy of dry electrodes, and the scalability of virtual environments should be improved in terms of covered space.

Locomotion techniques have important effects on spatial cognition in a virtual environment [51,52]. Arm-cycling has been proven to enable high spatial awareness [53]. On the other hand, teleportation can produce spatial disorientation due to a lack of self-motion cues [54,55] and more directional errors than continuous locomotion techniques [56]. Despite these issues, distance estimation, which is important for the assessment of spatial relations [57], is more accurate for teleporting than for artificial locomotion [58].

### 6.2. Analysis of Spatial Presence Data

A subset of items from the *MEC-SPQ* questionnaire was selected to collect feedback from 10 users who tested the application. Taking into account the mental disability suffered by the trial users, only the factors *Attention allocation*, *Spatial situation model*, *Spatial Presence—Self Location*, *Spatial Presence—Possible Actions*, and *Higher Cognitive Involvement* were retained in the reduced and simplified version of the questionnaire. For each factor, the authors of the MEC questionnaire proposed three possible scales consisting of 8, 6, and 4 items, respectively. For each of the factors included in this study, the scale consisting of only 4 items was adopted in order to make the questionnaire as simple as possible. For each item, users were asked to express their opinion on a 5-point Likert scale ranging from 1 ("I do not agree at all") to 5 ("I fully agree"). Scores on a scale from 0 to 4 were obtained by subtracting 1 from the levels selected by the users in response to the questionnaire items.

Table 1 shows the items of the questionnaire administered to users and the factors to which they belong.

Table 2 shows the average values, standard deviations, and coefficients of variation (i.e., the ratios between means and standard deviations) of the factors calculated over the scores expressed by the 10 users involved in the application test.

**Table 1.** Short version of the MEC Spatial Presence Questionnaire.

| Factor | Item |
|---|---|
| Attention allocation | I devoted my whole attention to the application |
| | I concentrated on the application |
| | The application captured my senses |
| | I dedicated myself completely to the application |
| Spatial situation model | I was able to imagine the arrangement of the spaces presented in the application very well |
| | I had a precise idea of the spatial surroundings presented in the application |
| | I was able to make a good estimate of the size of the presented space |
| | Even now, I still have a concrete mental image of the spatial environment |
| Spatial presence: self location | I felt like I was actually there in the environment of the presentation |
| | It was as though my true location had shifted into the environment in the presentation |
| | I felt as though I was physically present in the environment of the presentation |
| | It seemed as though I actually took part in the action of the presentation |
| Spatial presence: possible actions | I had the impression that I could be active in the environment of the presentation |
| | I felt like I could move around among the objects in the presentation |
| | The objects in the presentation gave me the feeling that I could do things with them |
| | It seemed to me that I could do whatever I wanted in the environment of the presentation |
| Higher cognitive involvement | I thought most about things having to do with the application |
| | I thoroughly considered what the things in the presentation had to do with one another |
| | The application presentation activated my thinking |
| | I thought about whether the application presentation could be of use to me |

**Table 2.** Mean scores, standard deviations and coefficients of variation of the user experience factors.

| Factor | Mean Score | Standard Deviation | Coefficient of Variation |
|---|---|---|---|
| Attention allocation | 3.900 | 0.166 | 0.043 |
| Spatial situation model | 3.625 | 0.422 | 0.116 |
| Spatial presence: self location | 3.850 | 0.320 | 0.083 |
| Spatial presence: possible actions | 3.350 | 0.502 | 0.150 |
| Higher cognitive involvement | 3.700 | 0.245 | 0.066 |

The mean scores and standard deviations of the factors were represented graphically in the histograms in Figure 7. There are no particularly marked differences between the average scores of the considered user experience factors. However, the factor with the highest average scores is *Attention allocation*, while the factor with lowest scores is *Spatial presence—possible actions*, despite the high values of *Spatial presence—self location* that make it the second highest scoring factor. In addition, the very low coefficient of variation indicates a strong agreement between users' opinions regarding *Attention allocation*. The average values of *Spatial situation model* are halfway between the two components of *Spatial presence*, as both have an influence on the mental model.

*Spatial presence—possible actions* and *Spatial situation model* have the highest coefficients of variation, which indicate some variability among users' opinions.

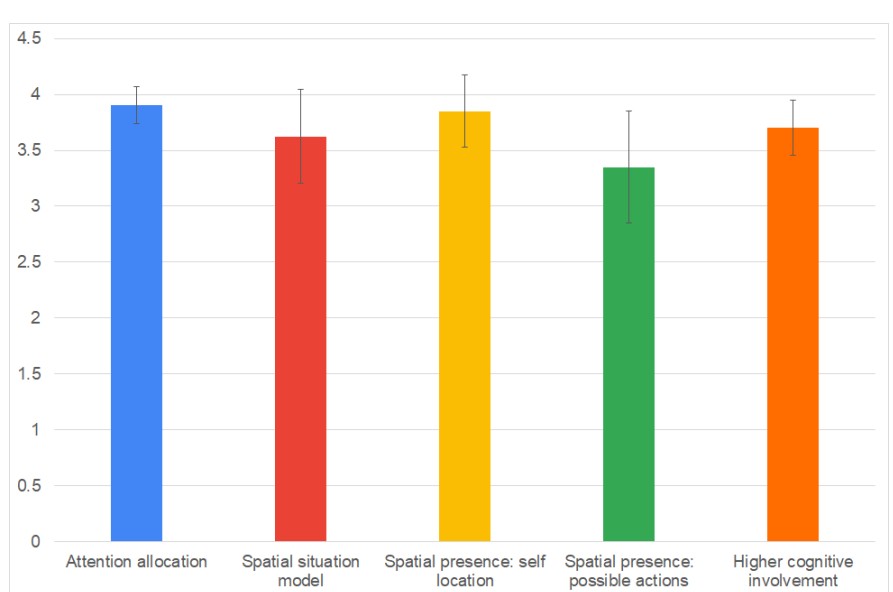

**Figure 7.** Histogram comparing the scores of user experience factors.

The density ridgeline plot in Figure 8 compares the distributions of the scores based on the density estimates of the factors calculated for the users involved in the tests. Black spots represent the original data points from which the distributions are generated.

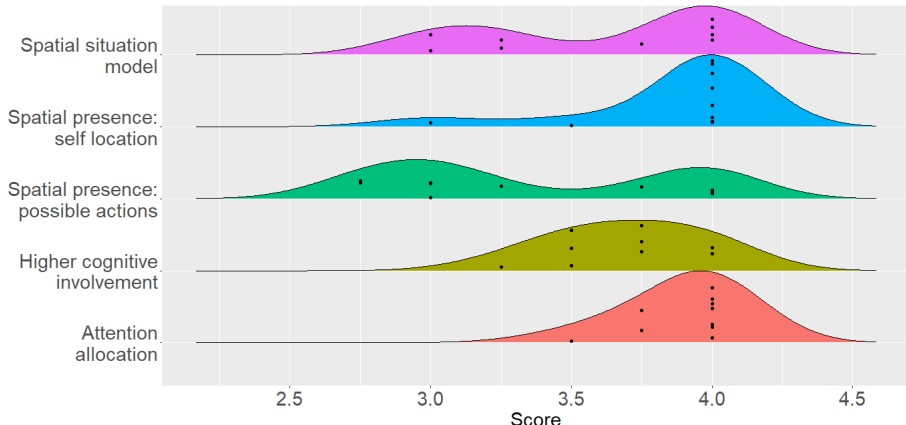

**Figure 8.** Distributions of factor scores.

The *Attention allocation* waveform has a very pronounced peak at almost the maximum score of 4.

The *Spatial presence—possible actions* waveform shows two clusters of scores: the most populated of the two seems to be the one consisting of the lowest scores, which are distributed around a value of 3. On the contrary, the waveform of *Spatial presence—self location* has a very pronounced promontory at a value of just under 4, indicating that almost all users rated it highly. The *Spatial situation model* waveform can almost be considered the result of the combination of these two factors: it has two promontories, of which the slightly more pronounced one almost corresponds to the peak of *Spatial presence—self location*.

The *Higher cognitive involvement* waveform has no real peak, but rather a rather broad hump, suggesting variability in a rather wide range and the absence of particular clusters of scores.

These observations thus suggest that, despite the good level of attention and involvement, also generated by the possibility of users to recognize their own works of art in the virtual museum, the users seem to be divided into two groups, one of which is somewhat more confused about the actions they can perform. Despite this difference in the mental model of the two groups of users, they seem to perceive the mere sense of embodiment, i.e., the feeling of being in the virtual environment, in the same way. Both locomotion and teleportation implemented in the application do not seem to have particularly affected this component. Sometimes an operator had to intervene to manage such modes of user movement via the controller: this does not seem to have caused disorientation in terms of embodiment, but may have been the cause of the confusion perceived by some users about the actions that can be performed.

### 6.3. Well-Being Assessment

The level of user well-being was assessed by means of a simplified questionnaire inspired by the UCL Museum well-being Measures Toolkit [59]. For each item, users were asked to express their opinion on a 5-point Likert scale ranging from 1 ("I do not agree at all") to 5 ("I fully agree") after the whole experience. Scores on a scale from 0 to 4 were obtained by subtracting 1 from the levels selected by the users in response to the questionnaire items.

Table 3 shows the average values, standard deviations, and coefficients of variation for each well-being item. With the exception of only one user who gave a less than full score for group work and interaction with others, all users gave the maximum score for all questionnaire items.

**Table 3.** Mean scores, standard deviations and coefficients of variation of the items of the simplified well-being questionnaire.

| Item | Mean Score | Standard Deviation | Coefficient of Variation |
|---|---|---|---|
| I felt happy | 4 | 0 | 0 |
| I felt involved | 4 | 0 | 0 |
| I felt at ease | 4 | 0 | 0 |
| I felt safe | 4 | 0 | 0 |
| I enjoyed working in a group | 3.9 | 0.3 | 0.077 |
| I interacted with others | 3.9 | 0.3 | 0.077 |

## 7. Conclusions

The current work aimed to illustrate a social inclusion project through the creation of "workshops", i.e., safe places in which to carry out activities based on the combination of art and new technologies. Fundamental to this was the implementation of an inclusion strategy, aimed at the recognition and enhancement of each person's differences, with a perspective of teamwork based on collaboration and continuous confrontation between participants, educators and experts in cultural heritage and technologies.

The beneficial exchange, which is the basis of the interaction between all those involved, made it possible to pursue inclusive education with the aim of ensuring full freedom of expression and equal educational opportunities for those involved, in a fair exchange.

The choice to introduce participants to new technologies, preferring gamification-based teaching with simple and clear language, was well received by all participants, demonstrating that, if well structured and calibrated to each person's needs, new technologies can greatly reduce the limitations that the real world, despite everything, still imposes. Experimental tests measured the impact on participants' well-being and the user experience related to the Virtual Museum experience, which nonetheless needs further upgrades to make the experience smoother, supporting the state of well-being.

The results of experimental tests on a sample of users revealed a high level of attention and involvement during the experience. From the point of view of spatial presence, the data showed a good general level of self-embodiment in the virtual environment but, at the same time, also the difficulty of some users in imagining the actions they could perform within it.

The goals achieved in terms of well-being can be attributed to increased levels of participation, inclusion, self-awareness, self-esteem, and social skills.

The project has established the basis for the implementation of methods and practices in and outside of workshop contexts in order to perceive diversity as an enrichment for the whole community.

**Author Contributions:** Conceptualization, S.C., F.F., G.S., S.L., L.C., C.G., V.D.L. and L.T.D.P.; methodology, S.C., F.F., G.S., S.L., L.C., V.D.L., C.G. and L.T.D.P.; software, S.C., F.F., G.S., S.L., L.C. and C.G.; technological investigation, S.C., F.F., G.S., S.L., L.C., C.G., V.D.L. and L.T.D.P.; validation, V.D.L. and L.T.D.P.; data curation, V.D.L. and C.G.; formal analysis, V.D.L.; visualization, V.D.L.; writing—original draft, S.C., F.F., G.S., S.L., L.C., C.G. and V.D.L.; writing—review and editing, S.C., F.F., G.S., S.L., L.C., C.G., V.D.L. and L.T.D.P. All authors have read and agreed to the published version of the manuscript.

**Funding:** "Includiamoci" project was funded under the public notice "Giovani per il sociale 2018", intended for social inclusion and personal growth, and promoted by the Department for Youth Policy and Universal Civil Service (Presidency of the Council of Ministers).

**Data Availability Statement:** Not applicable.

**Acknowledgments:** The authors would like to thank the "NovaVita Elena Fattizzo" Association (Casarano, Lecce).

**Conflicts of Interest:** The authors declare no conflict of interest.

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
