# Peer review of "Virtual Reality and Spatial Augmented Reality for Social Inclusion: The “Includiamoci” Project"

_information, doi:10.3390/info14010038_

Round 1

Reviewer 1 Report

The manuscript is dedicated to the XR-based Includiamoci project. The authors name the intention of this paper as follows: “ The project, which was briefly introduced in an earlier preliminary paper [ 1], is presented here in a more extended form that includes more technical details on application development (in particular on the implementation of a video mapping experience for the theater) as well as a user experience study.“ (ll.25-29). In general, I highly appreciate the project and its applicational intention. There are just a few minor points (to be seen as suggestions rather than mandatory points) I would like to mention:

1)      The user study reported in this manuscript is included without a smooth introduction with reference to existing user studies. Aspects of user cognition (see point above) are also missing when introducing the purpose of this study. Could you better derive from existing literature why this study was needed and what you expected from it (hypotheses)?

2)      You hardly refer to the important aspect of spatial cognition in XR applications. Using XR-based technologies as media transporting spatial information, requires onsidering the cognitive processing of spatial information in these media. In the last years, research on these aspects has produced several new findings. Your manuscript would benefit if such studies would be introduced in an individual section and also referred to in a discussion of your empirical results. Some examples (many more exist) are:

https://doi.org/10.1038/s41598-018-29029-x

https://doi.org/10.3390/ijgi10030150

https://doi.org/10.3389/fnins.2019.01439

Your only reference to spatial aspects so far is your reference to the Spatial Situation Model: “The Spatial Situation Model, a precondition for Spatial Presence, is a mental model [24] of the spatial environment based on the perceived spatial cues and on a user’s spatial memories and cognitions [25].“ (ll. 587-589).

Author Response

1) The user study reported in this manuscript is included without a smooth introduction with reference to existing user studies. Aspects of user cognition (see point above) are also missing when introducing the purpose of this study. Could you better derive from existing literature why this study was needed and what you expected from it (hypotheses)?

At the end of the 'Related work' section, we have tried to better explain the rationale behind the project presented in the paper, linking up with other works that have dealt with Spatial Learning, Spatial Presence and Spatial Memory.

2) You hardly refer to the important aspect of spatial cognition in XR applications. Using XR-based technologies as media transporting spatial information, requires considering the cognitive processing of spatial information in these media. In the last years, research on these aspects has produced several new findings. Your manuscript would benefit if such studies would be introduced in an individual section and also referred to in a discussion of your empirical results. Some examples (many more exist) are:

  1. https://doi.org/10.1038/s41598-018-29029-x
  2. https://doi.org/10.3390/ijgi10030150
  3. https://doi.org/10.3389/fnins.2019.01439

We have divided the section "User experience evaluation" into three subsections.

In the subsection "Spatial Presence model" in particular, we have expanded on the concept of Spatial Presence and the related model, also adding citations to more recent works (including those suggested by the reviewer).

In the subsection "Analysis of spatial presence data", in which we reported and discussed the results of the Spatial Presence questionnaire, we also referred to the two modes of user movement (teleportation and locomotion) referred to in the subsection "Spatial Presence Model" and both implemented in the virtual environment.

Finally, the subsection "Well-being assessment" reports the results of the well-being questionnaire.

Reviewer 2 Report

The paper is well-written and discusses an interesting theme. I recommend it for publication, although some minor aspects of the text could be improved:

1) Section 3.4: Provide a definition of AR/VR and MR, and explain in a few words what they are:

J. Steuer, ‘Defining Virtual Reality: Dimensions Determining Telepresence’, Journal of Communication, vol. 42, no. 4, pp. 73–93, 1992.

G. Cattan and C. Mendoza, ‘Virtual Reality: Definition and Craze’, IHMTEK ; IBM, Other, Oct. 2021. Accessed: Dec. 02, 2022. [Online]. Available: https://hal.archives-ouvertes.fr/hal-03361952

2) The legend of Figures 1, 2, and 3 seem misaligned.

3) Product company and country should be added. E.g.:

Unity 3D (Unity, San Francisco, CA, the US).

4) Section 4.  Provide an introduction to Unity before mentioning it, e.g:

"Unity 3D (...) is a popular engine for game and application development. While the code is written in C#, Unity 3D is able to build for multiple platforms including a wide range of MR devices. For this reason, we choose it as .... Concurrent platforms such Unreal Engine (...) was considered but..."

Author Response

1) Section 3.4: Provide a definition of AR/VR and MR, and explain in a few words what they are:

  • Steuer, ‘Defining Virtual Reality: Dimensions Determining Telepresence’, Journal of Communication, vol. 42, no. 4, pp. 73–93, 1992.
  • Cattan and C. Mendoza, ‘Virtual Reality: Definition and Craze’, IHMTEK ; IBM, Other, Oct. 2021. Accessed: Dec. 02, 2022. [Online]. Available: https://hal.archives-ouvertes.fr/hal-03361952

We have added clearer definitions of the concepts of VR, AR and MR, including a citation to the following comprehensive and all-encompassing book:

  • Alcañiz, M.; Sacco, M.; Tromp, J. Roadmapping Extended Reality: Fundamentals and Applications; Wiley, 2022

2) The legend of Figures 1, 2, and 3 seem misaligned.

We improved the figure captions and their positioning. In order to make the work better understood, we have added a new figure 1 with the caption 'The Art Therapy Lab' and a new figure 5 with the caption 'Virtual museum navigation in Unity'. We have deleted the previous figures 2 and 4.

3) Product company and country should be added.

We have added the requested information.

4) Section 4. Provide an introduction to Unity before mentioning it.

We have added an introduction to Unity.
